# Labelling affects agreement with political statements of right-wing populist parties

**Henrike Neumann[1], Isabel Thielmann[2], Stefan Pfattheicher[1] ***

**1** Aarhus University, Aarhus, Denmark, **2** University of Koblenz-Landau, Mainz, Germany

\* sp@psy.au.dk

## Abstract

In light of the recent rise of right-wing populist parties across Europe, it is an intriguing question under which conditions people agree with right-wing political statements. The present study investigates whether *mere labelling* of political statements as endorsed by a right-wing populist party influences people's agreement with such statements. In the study (pre-registered; $N = 221$ German voters), it is shown than that supporters of the right-wing populist party indicated higher agreement with the statements when they were labelled as being endorsed by the party (vs. not labelled), whereas non-supporters indicated lower agreement with the labelled than with the non-labelled statements. We conclude that labelling of the *very same* political statements changes voters' agreement with these statements. The results imply that rather than (dis)agreeing with the content of the statements per se, people may (dis)agree with right-wing populist statements because they come from a specific source (i.e., the right-wing populist party).

## Introduction

In recent years, right-wing populist parties have noticeably been on the rise in countries across Europe, significantly increasing their vote share in many national and regional elections [1, 2]. In ongoing debates, both scholars and politicians discuss why voters might agree with statements coming from right-wing populist parties. Here, we investigate whether *explicitly labelling* political statements that were endorsed by a right-wing populist party as coming from this party or not affects individuals' agreement with these statements. We thus examine whether agreement with the *very same* statement changes when a right-wing populist party label is affixed to it (i.e., varying the source information) and whether individuals' political orientation moderates corresponding labelling effects on agreement.

The effect of labelling has been demonstrated for a wide variety of stimuli. For example, when the very same apple juice was labelled as coming from a local farmer vs. a big brand manufacturer, participants rated the taste of the apple juice better [3]. In fact, effects of labelling have been found for the labelling of common familiar objects [4], spatial scenes [5], and faces [6], but also for modalities such as color [7], speed [8], taste [9], and odor [10]. Effects have also been found for labelling of more abstract concepts, such as the diagnostic labelling of mental disorders in court hearings [11]. In general, it appears that attaching labels to a

**Data Availability Statement:** All data and study materials are available online on the Open Science Framework (see https://osf.io/zu6td/?view_only=9a56f455f938452695f8524fbb7fb29c).

**Funding:** The authors received no specific funding for this work.

**Competing interests:** The authors have declared that no competing interests exist.

stimulus can interfere with how the stimulus is perceived or judged: When a stimulus is labelled, the perception more closely resembles the label rather than the original stimulus [12].

In the present work, we test the effect of labelling in the context of political statements. In fact, showing that agreement with the very same political statement changes depending on its label would help better understand why people agree with certain statements. Specifically, it is plausible to assume that agreement is not only driven by the content of certain statements, but also by how the statements are labelled in terms of the party from which they originate (i.e., the source). We therefore combine research on labelling with the timely issue of right-wing populism in the present study.

Transferring previous findings of labelling effects to the issue of political statements, it is to be expected that such statements are perceived differently if a certain label is affixed to them, meaning that a labelling effect would occur. Similar to the effects observed with regard to other stimuli, attaching a label to political statements may distort how the statements are perceived and judged, in the sense that the statements might be perceived more in accordance with the affixed label as compared to when no label is present. Specifically, we tested the hypothesis that supporters of a right-wing populist party in Germany show higher agreement with political statements endorsed by the party when they are labelled (vs. not labelled) correspondingly; conversely, non-supporters of right-wing populist parties should show lower agreement with the statements when they are labelled (vs. not labelled) as endorsed by such a party. Statistically speaking, we thus tested whether labelling a political statement as being endorsed by a right-wing populist party interacts with the initial support for this party in predicting agreement with the statements.

## Study

### Methods

**Research ethics statement and data.** The present study was conducted in full accordance with the Ethical Guidelines of the American Psychological Association (APA). Institutional review boards or committees are not mandatory in the country where the investigator responsible for the present study (last author) is employed (Denmark). Participants provided written informed consent prior to participating in the study. There was no deception of participants. All data and study materials are available online on the Open Science Framework (see https://osf.io/zu6td/). The study was pre-registered prior to the start of data collection (see https://aspredicted.org/mc6gb.pdf)

**Initial agreement with political parties.** As initial measurement, participants were asked to indicate their overall agreement with the general standpoints of the six largest political parties in Germany. Responses were given on a five-point Likert-type scale ranging from 1 = *no agreement* to 5 = *very strong agreement*. The parties—and the level of participants' support for them—were as follows: CDU/CSU (Christian Democrats; $M = 2.79$, $SD = 1.10$); SPD (Social Democrats; $M = 3.00$, $SD = 1.04$); Bündnis 90/Die Grünen (Green Party; $M = 2.89$, $SD = 1.18$); Die Linke (Left; $M = 2.73$, $SD = 1.36$); FDP (Liberal Party; $M = 2.55$, $SD = 1.12$); and AfD (right-wing populist party "Alternative for Germany"; $M = 1.88$, $SD = 1.26$).

**Agreement with political statements.** To assess our dependent variable, participants were presented with 16 political statements and asked to indicate the degree to which they agree or disagree with each. The statements were taken from the German voting advice application *Wahl-O-Mat* for the federal elections in 2017. In this application, users indicate their agreement with a variety of political statements and, based on parties' agreement or disagreement with these statements, learn which parties' program overlaps most with their own opinion. For the current study, we specifically selected *Wahl-O-Mat* statements that were endorsed

by the right-wing populist party *Alternative für Deutschland* (AfD). Sample items read, "There should be a limit to the admission of asylum seekers per year"; "Germany's defense expenditure should be increased"; and "Germany should return to a national currency" (translated from German, which was the original language of the study.) All items can be found on the OSF (https://osf.io/zu6td/). Responses were given on a five-point Likert scale ranging from 1 = *strongly disagree* to 5 = *strongly agree* (M = 3.06, SD = 0.60).

**Conditions.** Participants were randomly assigned to one of two conditions. In the control condition (*n* = 111), the political statements did not receive any label, and participants merely indicated the extent to which they agreed or disagreed with the statements. In the experimental condition (*n* = 110), participants were presented with the same political statements and asked to indicate their agreement or disagreement; this time, though, the statements were labelled with the logo of the AfD and it was further explicitly noted that the statements were endorsed by the AfD.

*Participants.* We conducted an *a priori* power analysis using G*Power [13] to determine the sample size required to detect a small to medium-sized effect ($f^2$ = .085) with 90% power, resulting in a required *N* = 126. However, to allow detection of even smaller effect sizes with sufficient power, we pre-registered to collect data of up to 200 participants, if possible.

Participants were recruited via the reliable online platform *Clickworker* (a platform providing access to people who perform discrete on-demand tasks, for instance, completing questionnaires in exchange for money). In total, 221 participants completed the study (thus oversampling slightly due to the Clickworker system which opened the study for more than the requested 200). No data were excluded. Each participant was compensated with 0.50 Euro. Participants were heterogeneous with regard to gender (40.7% were women, 57.9% men, 1.4% unspecified) and age, which ranged from 18 to 69 years (*M* = 37.97, *SD* = 12.07), as well as education (including participants from lower secondary education to higher education levels with access to universities). All participants indicated that they were native German speakers who are allowed to vote in Germany.

## Results

We observed medium levels of agreement with the political statements (possible range: 1–5); across conditions the mean was 3.06 (*SD* = 0.60). Agreement with the political statements differed significantly between the two conditions in that participants showed less agreement in the experimental "labelling" condition (*M* = 2.97, *SD* = 0.68) than in the control condition (*M* = 3.20, *SD* = 0.49), *t*(219) = 2.98, *p* = .003, Cohen's *d* = 0.40.

To test the hypothesized interaction between labelling and political orientation on agreement with the political statements, we regressed the agreement with political statements on the initial support for the right-wing populist party (AfD), labelling condition (dummy-coded; 0 = no label, 1 = label), and their interaction. Supporting our hypothesis, there was a significant interaction between support of the right-wing populist party and the labelling of statements as endorsed by this party (see Table 1). That is, as depicted in Fig 1, the relation between initial support of the right-wing populist party and agreement with the political statements was stronger when statements were labelled as being endorsed by the party than when they were not.

Additional simple slope analyses indicated that those participants who showed initial antipathy towards the right-wing populist party (scoring 1.00 on the 1 to 5 party-support item) indicated significantly lower agreement with the statements when they were labelled as being endorsed by this party as compared to when the statements were not labelled as such. In contrast, participants showing high initial support for the right-wing populist party (scoring ≥ 4.77 on the 1 to 5 party-support item) indicated significantly higher agreement

**Table 1. Estimated agreement with the 16 political statements as a function of initial support of the right-wing populist party, experimental condition, and their interaction (N = 221).**

|  | B | SE | t | p | LLCI | ULCI |
|---|---|---|---|---|---|---|
| Constant | 2.95 | 0.08 | 35.30 | < .001 | 2.79 | 3.12 |
| Labelling condition | -0.63 | 0.12 | -5.14 | < .001 | -0.87 | -0.39 |
| Support of right-wing populist party | 0.14 | 0.04 | 3.67 | < .001 | 0.06 | 0.21 |
| Labelling condition × Support | 0.20 | 0.05 | 3.74 | < .001 | 0.10 | 0.31 |
| *Conditional effect at low support of the right-wing populist party* |  |  |  |  |  |  |
| Labelling condition | -0.43 | 0.08 | -5.15 | < .001 | -0.59 | -0.26 |
| *Conditional effect at medium support of the right-wing populist party* |  |  |  |  |  |  |
| Labelling condition | -0.02 | 0.09 | -0.26 | .797 | -0.20 | 0.15 |
| *Conditional effect at high support of the right-wing populist party* |  |  |  |  |  |  |
| Labelling condition | 0.38 | 0.18 | 2.10 | .037 | 0.02 | 0.74 |

*Note.* The experimental condition (labelling) is coded 1, the control condition (no labelling) is coded 0; low [medium/high] support of the right-wing populist party refers to the estimated effect at 1.00 [3.00/5.00] on the 1–5 party-support item; CIs refer to the 95% confidence interval.

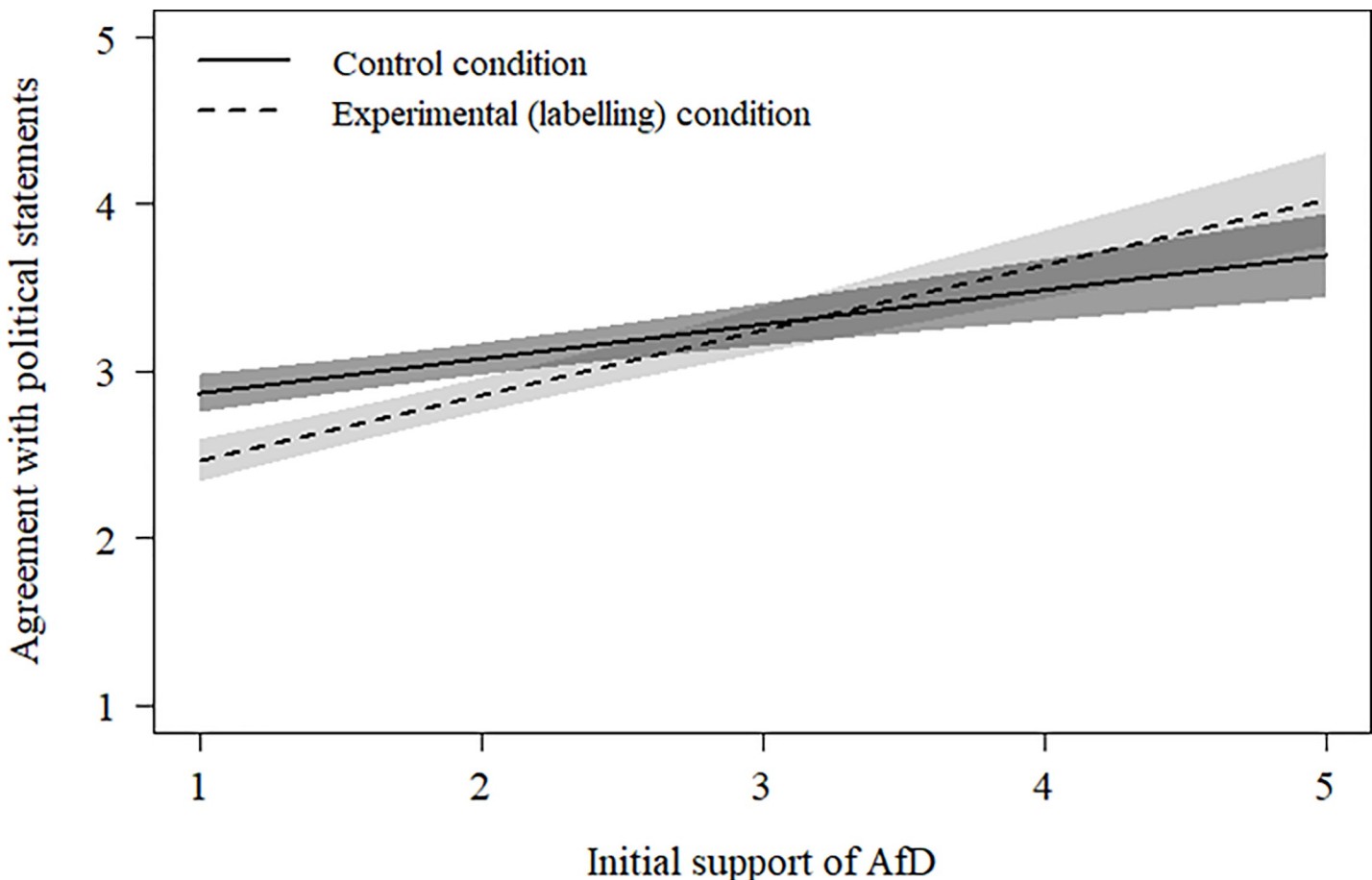

**Fig 1. Estimated agreement with the political statements as a function of initial support of the right-wing populist party (AfD) and whether or not the statements were labelled (N = 221); the shaded areas represent 95% confidence intervals.**

with the statements when they were labelled (vs. not labelled) as being endorsed by this party. This latter effect reached a conventional level of statistical significance only at the upper end of initial support for the right-wing populist party (see 95% CIs in Fig 1).

In additional exploratory analyses, we further regressed, in separate regressions, the agreement with political statements on the initial support for any other (non-right-wing) political party, labelling condition, and their interaction. None of the interactions turned out significant (all $p$s > .18). These findings indicate that the labelling effect specifically occurred for the (non-)supporters of the right-wing populist party, but not for supporters of other parties more generally.

## Discussion

Across Europe, right-wing populist parties are increasingly gaining acceptance. In the present study, we aimed at examining whether labelling political statements as endorsed by a right-wing populist party affects how these statements are perceived by supporters vs. non-supporters of the party. Overall, we tested the hypothesis that labelling political statements as being endorsed by a right-wing populist party increases agreement with the statements among supporters of the right-wing party, but decreases agreement among supporters of other, non-right-wing parties.

Results generally supported our hypothesis, showing that labelling of political statements does indeed have an effect on individuals' agreement or disagreement with these statements. Specifically, we found differential effects for supporters of the AfD and supporters of other parties: Supporters indicated higher agreement with the statements when they were labelled (vs. not labelled) as being endorsed by the AfD, whereas non-supporters showed lower agreement with the labelled (vs. not labelled) statements.

As such, the present work stands in the tradition of other research showing that a message will be received very differently depending on what source the message is believed to come from. For example, Cohen [14] showed that liberal U.S. participants were more likely to approve a policy that was allegedly supported by the Democratic (vs. Republican) party whereas conservative participants were likely to approve a policy that was allegedly supported by the Republican (vs. Democratic) party. Moreover, our findings are in line with research on intuitive decision making (e.g., [15]), which proposes that individuals have an initial intuitive response towards information provided (e.g., a political statement, or a case of a moral violation). Applied to the present study, it is conceivable that non-supporters of the right-wing populist party have a negative intuitive response towards political statements when they are labelled as being endorsed by a right-wing populist party, whereas supporters of right-wing populist parties may have a positive intuitive response towards such statements when they are labelled as being endorsed by a right-wing populist party.

### Theoretical implications, limitations, and directions for future research

As our results show, labelling of political statements can change the way such statements are evaluated. There are at least two mechanisms through which labelling could influence individuals' perception of political statements. First, the label may change the statements' inherent meaning. When labelled as being endorsed by a right-wing populist party, the understanding of a political statement may be different as compared to when it is labelled as being endorsed by more mainstream parties or, as in this study, if it does not carry any label. That is, seemingly general statements may come to have a more specific and perhaps more extreme meaning when endorsed by a right-wing populist party. For instance, a statement on restricting the migrant flow into Germany (e.g., "The migrant flow into Germany should be limited.") may,

when being labelled as coming from a right-wing populist party, be perceived to imply the introduction of specific restrictive measures and likely a rather low number of migrants to be tolerated. By contrast, the very same statement may convey a more tolerant meaning when it comes from other, non-right-wing parties. Understanding the statement's meaning in such a different way when it carries the right-wing party label may be appealing to right-wing party supporters, whereas it may repel non-right-wing party supporters.

Second, the label of endorsement conveys knowledge about the source of the respective statements, and this knowledge itself may automatically change individuals' judgement of the statements. That is, regardless of the statements' precise content or meaning, perceiving the right-wing populist party to be the source may lead non-supporters of the party to disagree with the statement while leading supporters of the party to agree with it. This could be due to genuine antipathy towards, or affinity for, the right-wing populist party, as well as to certain implications that this knowledge bears for individuals, for instance, with regard to social desirability and societal acceptance.

Based merely on the data collected in the current study, it is impossible to determine which of these two mechanisms can account for the observed labelling effect, that is, whether the labelling changes the inherent meaning of the statements or whether knowledge about the source itself changes individuals' evaluation of the statements without affecting the meaning in itself. It may also be a mixture of both mechanisms, rendering it even more difficult to disentangle the two. Future research is needed to illuminate the exact cause of the observed labelling effect.

As an additional limitation, it should be noted that the current study was conducted in Germany and therefore referred to the German right-wing populist party AfD, using only statements from a German voting advice application. Thus, caution is warranted in generalizing the findings and inconsiderately transferring them to other countries and political systems or parties, given that the situation in every individual country may be unique. However, considering the fact that there is rising support for right-wing populism across Western countries in general (e.g., [1])—and also in light of the impressive consistency of labelling effects [12]—we expect the results to hold in other countries as well. Nonetheless, future research should attempt to replicate the present findings in different countries and settings.

The present research can further be discussed from a social identity perspective (e.g., [16]). Based on social identity theory, it has been argued that a shared social identity of a sender and receiver of a message influences the persuasiveness of the message (e.g., [14, 17]). This idea has since received empirical support in experimental [18] and qualitative [19] research on populist political messaging. Applied to the present study, it is plausible that the logo of the right-wing populist party has activated supporters' subjective categorization and sense of belonging to the group of right-wing party supporters. Thus, the findings might be traced back to an expression of ingroup favoritism manifested in increased positive evaluation of the political statements when labelled correspondingly. Conversely, the logo of the right-wing populist party may have also activated non-supporters' subjective categorization and sense of belonging to the group of opponents to right-wing party supporters. As a downstream consequence, these individuals might have engaged in outgroup derogation manifested in decreased positive evaluation of the political statements when labelled as coming from the right-wing populist party. However, these are all post-hoc speculations that need further empirical investigation.

## Societal implications

The present study has implications for politics and society with regard to right-wing populism. It appears that non-supporters of right-wing populist parties do not necessarily disagree with

statements made by these parties *per se*. Rather, disagreement with the parties' statements does, to some extent, result from the fact that the statements are endorsed by the party and not from actual disagreement with the content of the statements. Indeed, we observed medium levels of agreement with the political statements when the statements were not explicitly labelled as coming from the right-wing populist party (i.e., in the control group). People who do not usually support a right-wing populist party may not always oppose the very content endorsed by them, but they do appear to be repelled by the label indicating endorsement by such a party. This raises questions as to whether agreement with (extreme) political positions depends, to a significant extent, on their source and on individuals' attitudes towards that source.

Conversely, supporters of right-wing populist parties agree less with certain statements if they do not know that the statements stem from these parties. This suggests that rather than actual belief in the party's program, people may merely support and vote for right-wing populist parties in an effort to protest against established ones (e.g., [20]), and for reasons of social identity [17]. This, in combination with the generally rising support for right-wing populist parties across European countries, implies that established democratic parties at least partially fail to engage voters and to show satisfying responses to issues that some voters deem important.

Notwithstanding the importance of these potential societal implications, a critical note about what experiments—like the one presented in this contribution—can tell us about real-life contexts and phenomena seems in order (see also, e.g., [21]). With the present experimental research, we were able to examine our main hypothesis in a controlled environment, testing whether individuals' agreement with political statements depend on labelling of the statements' source. In this regard, we applied an analysis on the level of the individual (reflecting "methodological individualism", [22]). This perspective, however, neglects the dynamic interplay between the individual and the societal/macro level. Therefore, the direct projection of our findings onto real-life phenomena such as the dynamic rise and fall of right-wing populist parties in societies might be limited. That said, it is our hope that the present findings will encourage future research to also study the observed phenomena in a more applied setting while also taking the societal/macro level into account.

## Conclusion

The present research set out to assess whether labelling of political statements as endorsed by a right-wing populist party changes individuals' agreement with said statements in comparison to when no label is affixed to such statements—in this regard contributing to research on the persuasiveness of political messages as a function of source. We conclude that labelling of political statements matters and helps explain voters' (dis)agreement with political statements beyond the mere content of corresponding statements.

## Author Contributions

**Conceptualization:** Stefan Pfattheicher.

**Data curation:** Stefan Pfattheicher.

**Investigation:** Stefan Pfattheicher.

**Methodology:** Stefan Pfattheicher.

**Project administration:** Stefan Pfattheicher.

**Supervision:** Stefan Pfattheicher.

**Writing – original draft:** Henrike Neumann, Isabel Thielmann, Stefan Pfattheicher.

**Writing – review & editing:** Henrike Neumann, Isabel Thielmann, Stefan Pfattheicher.

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
