## [Decision Letter · Decision Letter 0]

17 Jul 2020

PONE-D-20-15712

Labelling affects agreement with political statements of right-wing populist parties

PLOS ONE

Dear Dr. Pfattheicher,

Thank you for submitting your manuscript to PLOS ONE. After careful consideration, we feel that it has merit but does not fully meet PLOS ONE’s publication criteria as it currently stands. Therefore, we invite you to submit a revised version of the manuscript that addresses the points raised during the review process.

We look forward to receiving your revised manuscript.

Kind regards,

Shang E. Ha, Ph.D.

Academic Editor

PLOS ONE

Journal Requirements:

2. Please provide additional details regarding participant consent.

In the Methods section, please ensure that you have specified (i) whether consent was informed and (ii) what type you obtained (for instance, written or verbal).

If your study included minors, state whether you obtained consent from parents or guardians.

If the need for consent was waived by the ethics committee, please include this information.

Reviewers' comments:

Reviewer's Responses to Questions

**Comments to the Author**

1. Is the manuscript technically sound, and do the data support the conclusions?

Reviewer #1: Yes

Reviewer #2: Yes

2. Has the statistical analysis been performed appropriately and rigorously? 

Reviewer #1: I Don't Know

Reviewer #2: Yes

3. Have the authors made all data underlying the findings in their manuscript fully available?

Reviewer #1: Yes

Reviewer #2: Yes

4. Is the manuscript presented in an intelligible fashion and written in standard English?

Reviewer #1: Yes

Reviewer #2: Yes

5. Review Comments to the Author

Reviewer #1: PONE Review MS PONE-D-20-15712

Labelling affects agreement with political statements of right-wing populist parties

This is an interesting and well-written short manuscript, one that reports the findings of a study examining whether ‘mere labelling political messages’ changes the way in which recipients receive the message in question.

More specifically, the study being reported examined whether people would respond differently (agree/disagree) to populist messages depending on whether or not the message was framed as originating from a populist party.

The findings reported in this paper are compelling and confirm that this was the case. And as far as I can judge this (as someone with modest expertise in regression analyses) the methods being deployed seem fine.

I very much like the paper’s main argument, and welcome this contribution because it fits nicely within a growing body of research in social and organizational psychology. That being said, there is also clear room for improvement, and in the remainder of this review I will list three issues that, in my view, would need to be addressed before this manuscript could be deemed suitable for publication in PLOS-One.

1. The authors claim ‘to be the first ones to examine the effect of labelling on political messages’. If we take this statement literal (‘labelling’) then it could be argued that this statement is correct. However, there are many other studies that have shown that the persuasiveness of a message depends on the (perceived) source, and when we take that into account, the statement is incorrect. This problem is easily fixed, and would merely involve refraining from making this, in my view inaccurate, statement.

2. Following on from the first point, the paper is under-referenced, and misses several highly relevant contributions already showing that a message will be received very differently depending on what source the message is believed to come from. For example, this idea is discussed in quite some detail in Jonathan Haidt’s 2012 book ‘The Righteous Mind’, in which Haidt recalls David Hume’s insight that “emotion and empathy precede reason”, (i.e. translated into plain English: we first lean one way or the other emotionally, and we subsequent retrofit reasons onto this emotion.

Social Identity Theorists (Tajfel & Turner, 1979 etc.) have explored this very phenomenon and advanced a compelling argument about the underlying process responsible for this mediating effect. One place where this is explained very clearly is John Turner’s 1991 book ‘Social Influence’, in which he argues on the basis of social identity theorizing, that ‘shared social identity’ mediates the persuasiveness of messages. This insight has since been demonstrated experimentally (Greenaway et al, 2015) and qualitatively in research into populist political messaging (Mols, 2012).

The idea that source information matters was also discussed in quite some detail in a recent paper in Evidence & Policy (Mols, Bell & Head, 2020). More specifically, in this paper it is argued that people’s receptiveness to ‘research evidence’ is mediated by shared social identity. This paper is useful because it considers real life scenarios (e.g. change climate skeptics who identify with a populist party and collectively despise climate researchers will continue to reject evidence of climate change, no matter how much additional research evidence will be presented to them).

Considering the above, it would be good to (a) refrain from claims about ‘being the first to study the effect of labeling on populist messages’, (b) avoid describing the study being reported as about ‘labelling’ and to describe it instead as a study examining whether source information mediates the persuasiveness of a populist message, (c) insert a literature review of studies examining the effect of source information on message persuasiveness (as per above), and (d) to mention ‘shared social identity’ as a likely factor in the section contemplating what could be the ‘underlying process’

3. In my view, the paper not only requires a proper literature review section (as per above suggestions), but also a section ‘limitations of the present research’. More specifically, it would be good if the authors could reflect on what experiments can tell us about real-life contexts. As Mols & ‘t Hart (2017) explain in their chapter ‘Political Psychology’, experimental research can be extremely useful for examining effects in a controlled environment (to test common assumptions and to debunk common myths). However, all too often researchers project the findings onto real-life phenomena without ever questioning the appropriateness of ‘methodological individualism’ (using individual-level findings to make claims about collective-level phenomena). It would in my view be good to explore such issues in a ‘limitations of the present research’ section, and to at least demonstrate a healthy level of awareness of these rather well-known methodological challenges and pitfalls.

In sum, in my view this is a promising paper, one that has the potential to make a valuable contribution to the ‘communication’ and ‘messaging’ literatures. However, as it stands the paper suffers from a number of shortcomings. None of these would seem fatal. Rather, in my view the issues I raised could be addressed relatively easily in a revision.

I wish the authors good luck with their revision, and I hope my suggestions will prove useful.

REFERERENCES

Haidt, J. (2012). The righteous mind: Why good people are divided by politics and religion. Vintage.

Greenaway, K. H., Wright, R. G., Willingham, J., Reynolds, K. J., & Haslam, S. A. (2015). Shared identity is key to effective communication. Personality and Social Psychology Bulletin, 41(2), 171-182.

Mols, F. (2012). What makes a frame persuasive? Lessons from social identity theory. Evidence & Policy: A Journal of Research, Debate and Practice, 8(3), 329-345.

Mols, F. & ‘t Hart (2018) Political Psychology. In: Vivien Lowndes, David Marsh & Gerry Stoker (eds.) Theory and Methods in Political Science, Fourth Edition. Palgrave-MacMillan.

Mols, F., Bell, J., & Head, B. (2020). Bridging the research‐policy gap: the importance of effective identity leadership and shared commitment. Evidence & Policy: A Journal of Research, Debate and Practice, 16(1), 145-163.

Turner, J. C. (1991). Social influence. Thomson Brooks/Cole Publishing Co.

Reviewer #2: The authors investigated whether labelling political statements as endorsed by a right-wing populist party influences people’s agreements with such statements. The research question is comprehensible and straight forward. The empirical part is neat and well done. The results are interesting and communicated in a clear and comprehensible way. I highly recommend publishing this paper. My only suggestion is to add a further potential explanation for the labelling effect. Social Identity Theory might predict the same result. The party label might trigger social identification and, related to this, make intergroup conflicts between parties and partisans more salient. Nice paper! Congrats.

6. PLOS authors have the option to publish the peer review history of their article (what does this mean?). If published, this will include your full peer review and any attached files.

Reviewer #1: No

Reviewer #2: No

---

## [Author Response · Author response to Decision Letter 0]

12 Aug 2020

Dear Editor Dr. Ha,

We submit here our revised manuscript titled “Labelling affects agreement with political statements of right-wing populist parties” (PONE-D-20-15712).

We appreciate the assistance and the constructive suggestions the reviewers provided. We have revised the manuscript accordingly and copied and pasted the revised or added text into our response letter. In the main manuscript, any revised or newly inserted text is featured in red highlight.

Thank you for your time and assistance with this process.

Kind regards, 

Henrike Neumann, Isabel Thielmann, and Stefan Pfattheicher

Contact information of the corresponding author:

Stefan Pfattheicher

Aarhus University

Department of Psychology and Behavioral Sciences

8000 Aarhus C, Denmark

sp@psy.au.dk

---

## [Decision Letter · Decision Letter 1]

14 Sep 2020

Labelling affects agreement with political statements of right-wing populist parties

PONE-D-20-15712R1

Dear Dr. Pfattheicher,

We’re pleased to inform you that your manuscript has been judged scientifically suitable for publication and will be formally accepted for publication once it meets all outstanding technical requirements.

Kind regards,

Shang E. Ha, Ph.D.

Academic Editor

PLOS ONE

Additional Editor Comments (optional):

Reviewers' comments:

Reviewer's Responses to Questions

**Comments to the Author**

1. If the authors have adequately addressed your comments raised in a previous round of review and you feel that this manuscript is now acceptable for publication, you may indicate that here to bypass the “Comments to the Author” section, enter your conflict of interest statement in the “Confidential to Editor” section, and submit your "Accept" recommendation.

Reviewer #1: All comments have been addressed

Reviewer #2: All comments have been addressed

2. Is the manuscript technically sound, and do the data support the conclusions?

Reviewer #1: Yes

Reviewer #2: Yes

3. Has the statistical analysis been performed appropriately and rigorously? 

Reviewer #1: Yes

Reviewer #2: Yes

4. Have the authors made all data underlying the findings in their manuscript fully available?

Reviewer #1: Yes

Reviewer #2: Yes

5. Is the manuscript presented in an intelligible fashion and written in standard English?

Reviewer #1: Yes

Reviewer #2: Yes

6. Review Comments to the Author

Reviewer #1: I am impressed with the way the authors have addressed my concerns, and I am happy to endorse this manuscript for publication in PLOS-ONE.

Reviewer #2: The authors addressed my comments adequately. I am happy with this revision. I can see that the theoretical part of this paper is extraordinarily short but I believe that the theoretical account of this paper is so straightforward and empirical research is so limited that an extensive literature review would just mean to link research that is not ultimately relevant for the present investigation. Thumbs up to the authors for this work.

7. PLOS authors have the option to publish the peer review history of their article (what does this mean?). If published, this will include your full peer review and any attached files.

Reviewer #1: No

Reviewer #2: No

---

## [Editor Report · Acceptance letter]

27 Oct 2020

PONE-D-20-15712R1 

Labelling affects agreement with political statements of right-wing populist parties 

Dear Dr. Pfattheicher:

I'm pleased to inform you that your manuscript has been deemed suitable for publication in PLOS ONE. Congratulations! Your manuscript is now with our production department. 

Kind regards, 

on behalf of

Dr. Shang E. Ha 

Academic Editor

PLOS ONE